

# LncRNA/miRNA/mRNA ceRNA network analysis in spinal cord injury rat with physical exercise therapy

Jiahuan Wu[1,*], Xiangzhe Li[1,*], Qinghua Wang[2], Sheng Wang[1], Wenhua He[3], Qinfeng Wu[1] and Chuanming Dong[3]

[1] Suzhou Science & Technology Town Hospital, Gusu School, Nanjing Medical University, Rehabilitation Medical Center, Suzhou, China
[2] Nantong University, Experimental Animal Center, Nantong, China
[3] Medical College of Nantong University, Department of Anatomy, Nantong, China
[*] These authors contributed equally to this work.

## ABSTRACT

Noncoding RNAs have been implicated in the pathophysiology of spinal cord injury (SCI), including cell death, glial scar formation, axonal collapse and demyelination, and inflammation. The evidence suggests that exercise therapy is just as effective as medical treatment in SCI. However, studies of competing endogenous RNA (ceRNA)-mediated regulation mechanisms in the therapy of SCI with exercise are rare. The focus of this research was to investigate the effect of exercise therapy on the expression levels of long noncoding RNA (lncRNA), microRNA (miRNA), and mRNA in rats with SCI. The RNA-seq technology has been used to examine the differentially expressed circRNAs (DECs), lncRNAs (DELs), miRNAs (DEMs), and genes (DEGs) between SCI and exercise therapy rats. The ceRNA network was established using interactions between miRNAs and mRNAs, as well as between miRNAs and lncRNAs/circRNAs. The Database for Annotation, Visualization, and Integrated Discovery was used to anticipate the underlying functions of mRNAs. Our current study identified 76 DELs, 33 DEMs, and 30 DEGs between groups of SCI rats and exercise therapy rats. Subsequently, these newly discovered ceRNA interaction axes could be important targets for the exercise treatment of SCI.

## INTRODUCTION

Neurological disorders such as traumatic spinal cord injury (SCI) and brain injury, central nervous system (CNS), stroke, neurodegenerative diseases, and malignancies are affecting an increasing number of people (*McDonald & Sadowsky, 2002*). Spinal cord injuries are among the most debilitating conditions on this list, as the affected individuals and their families are frequently deprived of attributes that permanently alter their life (*Carter, 2014*). Traffic accidents are the leading cause of SCI, followed by violent attacks, sports, falls, and industrial traumas (*Ahuja et al., 2017*; *Furlan et al., 2011*).

Corresponding authors
Qinfeng Wu,
wuqinfeng0911@163.com
Chuanming Dong,
yiyimarket@163.com

The white and grey matter of the spinal cord contains nerve cell bodies including ascending and descending tracts. As a result, the various locations and extents of SCI can result in variable degrees of disability, ranging from partial sensory or motor loss to full paralysis below the injured area, along with acute and chronic consequences (*Haisma et al., 2007*; *Adriaansen et al., 2013*). Currently, the most common treatment interventions for SCI are conservative surgery and medication (*Karsy & Hawryluk, 2019*; *Domurath & Kutzenberger, 2012*). However, owing to the loss of nerve tissue, the recovery ability of nerves is limited. For that reason, more research into the treatment and cure for SCI is required to establish more effective techniques for the prevention and treatment of this condition.

Physical activity and exercise are becoming increasingly important in the prevention and treatment of a variety of medical disorders (*Sharif et al., 2018*; *Booth, Roberts & Laye, 2012*). There is growing evidence that exercises benefits recovery of neuromuscular function from SCI (*Zbogar et al., 2016*; *Behrman, Ardolino & Harkema, 2017*). Physical exercise includig treadmill training is a frequently used approach for restoring the ability to walk after SCI. This extensive exposure of task-specific repetitive training helps promote reorganization of the primary motor cortex (*Ungerleider, Doyon & Karni, 2002*). Treadmill training significantly improves functional recovery and neural plasticity after incomplete spinal cord injury (*Sun et al., 2013*). Treadmill training significantly increased the expression of a neurotrophin brain-derived neurotrophic factor (BDNF) in the lumbar motoneurons as compared to non-training (*Wang et al., 2015*). However, the underlying definite explanation on the therapeutic role of Exercise is still mysterious. Noncoding RNAs, such as circular RNAs (circRNAs), long noncoding RNAs (lncRNAs), and microRNAs (miRNAs/miRs), have recently been discovered to have important roles in a variety of biological processes, including apoptosis and cell proliferation (*Lekka & Hall, 2018*; *Zhao et al., 2020*). The miRNAs act by binding to complementary sequences in the 3′-untranslated region (UTR) of their target mRNAs, preventing the transcript from being translated or causing it to be degraded (*Achkar, Cambiagno & Manavella, 2016*). Binding competitively to miRNAs along with their miRNA response regions, lncRNAs and circRNAs can perform as competing endogenous RNAs (ceRNAs), controlling the miRNA expression levels targeting mRNAs (*Panda, 2018*). So, the interaction of lncRNA/circRNA-miRNA-mRNA may be considered as a significant mechanism underlying the development and initiation of SCI. Previous research has supported this hypothesis (*Peng et al., 2020*; *Wang et al., 2020*). Certain miRNAs (miR-21, miR-486, miR-20) are dysregulated and associated with SCI through regulating inflammation, gliosis, cellular death, or regeneration, in microarray or sequencing investigations (*Nieto-Diaz et al., 2014*). We predicted that exercise might impact the lncRNA/circRNA-miRNA-mRNA interactions.

The focus of this research was to reveal the lncRNA-miRNA-mRNA ceRNA-mediated regulation pathways in exercise-treated rats as a preliminary approach. This study may provide the influences of such exercise therapies on the injured spinal cord.

## MATERIALS & METHODS

### Establishment of spinal cord injury models

We obtained all the adult female Sprague Dawley rats (220–250 g) from the Animal Care Facility of Nantong Medical University (Nantong, China). All animal experiments were performed in accordance with institutional norms and were permitted by Nanjing Medical University's Animal Care and Use Committee (S20200317-024). All animals were bred and housed (five per cage) in a controlled environment with an ambient temperature of $23 \pm 2\,°C$ and free access to water and food was given to animals for a 12-hour light-dark cycle. Rats received a hemisection SCI at the spinal level of Thoracic 10 as described in our previous report. The hemisection surgery was carried out by an expert in the SCI rat model to eliminate the error caused by the incision. Following surgery, an intramuscular injection of the Penicillin 20IU/d (i.m.) was given to the animals for 7 days after the operation. Rats were given bladder care three times each day for two weeks, or until the recovery of bladder control. Baytril (0.06 mg/kg) was injected subcutaneously two times per day for 7 days if there was any sign of infection. Certain criteria were considered for animals being included in or excluded from the experiment. The animal that met any two of the group I criteria ((a) rough coat and unkempt, (b) eyes completely or partially closed for 10 minutes, (c) markedly diminished resistance to being handled (grimace response), (d) markedly decreased movement/lethargy, (e) hunched posture, and (f) distended abdomen), were excluded and euthanized. Euthanasia was performed on mice who met one of the group II criteria ((a) inability to eat or drink, (b) moribund/unresponsive, (c) failure to right itself when placed on its back, (d) dyspnea, or (e) 15% or greater reduction in body weight). The rats were sacrificed two weeks after the second operation and the epicenter of the T10 spinal cord was collected for sequencing and q-PCR analysis. All mice were euthanized humanely, *via* carbon dioxide asphyxiation (all rats were included in the analysis).

### Exercise therapy administration

Animals were given body weight-supported Treadmill Training (TMT) for two weeks starting one week after SCI. The feet of the rats were secured to the pedals and suspended horizontally. The treadmill speed was kept at 6 m/min, and each exercise bout comprised two 20-minute workout intervals per day, five days per week for two weeks. The weight-supported range throughout the training was set at 20%–40% of the rat's weight, depending on the rat's functional state. Treatments were administered daily in the same arrangement and at the same time, and cages locations were kept at the same positions throughout the experiment to minimize confounders.

### Sample size for RNAseq studies

The RNA-seq study's error is caused by two factors: the sequencing's technical variability, and the biological variability of the specimens being compared within groups. For human samples, this value is frequently 4-1, but for inbred animal lines, it is frequently 1 or less. That is, a 2-fold difference is observed in mean expression between treated and untreated samples, while also seeing a 50% variation in expression within the control or treatment groups. This would correspond to an effect size of 2 and a biological coefficient of variation

in group 1 /or both (CV) of 0.5. In general, sequencing depths greater than $5/CV^2$ result in only minor increases in study efficiency and/or power, but adding more samples is always efficient.

Depth is a required argument; any one of the others may be left missing and the function will solve for it. An argument may be a vector, in case a vector of values is returned. If multiple arguments are vectors a matrix or array of results is returned. Values for alpha and power would be 0.05 and 0.9. The effect parameter specifies biological effect that, if it were true, the experimenter would want to be able to detect; commonly used. The statements that A has twice the expression of B has half of A are symmetric, values of 1.5 and 2 will the same result. The estimated CV of expression within group may be the most difficult parameter to choose the in depth discussion recommended values for types of data. Samples sizes in the two groups are equal ($n2 = n$) if a second sample is not . Likewise the coefficients of variation in the two groups are assumed to be equal if cv2 is not specified. We show the tables of sample size results at different depths according to each power value through https://git.bioconductor.org/packages/RNASeqPower (Table S1–S3).

## miRNA extraction and sequencing

In total, six samples (three from the SCI group and three from the SCI plus TMT group) were selected for high-throughput RNA sequencing. For each sample, three random rats spinal cord tissues in the same group were mixed together before RNA extraction. A five mm spinal cord was taken from the intact and injured site, the spinal cord segments were rapidly frozen on dry ice following the storage at $-80$ °C before RNA extraction. After total RNA was extracted by Trizol reagent kit (Invitrogen, Carlsbad, CA, USA), the RNA molecules in a size range of 18–30 nt were enriched by polyacrylamide gel electrophoresis(PAGE). After adding the 3′ adapters, the 36–44 nt RNAs were enriched. Additionally, the 5′ adapters were ligated to the RNAs. The ligation products were reverse transcribed using PCR amplification. The PCR products, which ranged in size from 140 to 160 bp, were enriched to create a cDNA library and sequenced on an Illumina HiSeqTM 2500 (Gene Denovo Biotechnology Co, Guangzhou, China). Results obtained from sequencing contained noise which would affect the following assembly and analysis. Thus, to get pure tags, raw reads were further filtered according to the fastqc software. Then, all purified tags were aligned with GeneBank, Rfan, and Genome. miRNA including existing miRNA, known miRNA, and novel miRNA was identified by software Mireap_v0.2. After tags were annotated as mentioned previously, the annotation results were determined in this priority order: rRNA etc > exist miRNA > exist miRNA edit > known miRNA > repeat > exon > novel miRNA > intron. The tags that cannot be annotated as any of the above molecules were recorded as unann. Correlation analysis of two parallel experiments provides the evaluation of the reliability of experimental results as well as operational stability. The correlation coefficient between two replicas was calculated to evaluate repeatability between samples. The closer the correlation coefficient gets to 1, the better the repeatability between two parallel experiments. We identified miRNAs with a fold change $\geq 2$ and $P$ value $<0.05$ in a comparison as significant DE miRNAs.

## mRNA and LncRNA bioinformatics analysis

Total RNA extraction was performed according to the manufacturer's procedure using the Trizol reagent kit (Invitrogen, Carlsbad, CA, USA) . The integrity of the RNA was determined using an Agilent 2100 Bioanalyzer (Agilent Technologies, Palo Alto, CA, USA) and validated using RNase free agarose gel electrophoresis. After extracting total RNA, rRNAs were removed to retain mRNAs and ncRNAs. The enriched mRNAs and ncRNAs were fragmented into short fragments by using fragmentation buffer and reverse transcribed into cDNA with random primers. Second-strand cDNA was synthesized by DNA polymerase I, RNase H, dNTP (dUTP instead of dTTP) and buffer. Next, the cDNA fragments were purified with QiaQuick PCR extraction kit (Qiagen, Venlo, the Netherlands), end repaired, poly(A) added, and ligated to Illumina sequencing adapters. Then UNG (Uracil-N-Glycosylase) was used to digest the second-strand cDNA. The digested products were selected on the bases of size by agarose gel electrophoresis, PCR product, and sequenced using Illumina HiSeqTM 4000 (or other platforms) by Gene Denovo Biotechnology Co. (Guangzhou, China). Clean reads were acquired using fastqc software (version 0.18.0) and reads were then mapped to the ribosomal RNA (rRNA) database using the short read alignment tool Bowtie2 (version 2.2.8). After that, the reads that had been mapped to rRNA were deleted. The remaining reads were further used in the assembly and analysis of the transcriptome. An index of the reference genome was built, and paired-end clean reads were mapped to the reference genome using HISAT2 (version 2.1.0). The reconstruction of transcripts was carried out with software Stringtie. To identify the new transcripts, all reconstructed transcripts were aligned to reference genome and were divided into twelve categories using Cuffcompare. Transcripts with one of the class codes "u,i,j,x,c,e,o" were defined as novel transcripts. Then we used the length of the transcript which was longer than 200bp and the exon number was more than 2 to identify reliable novel genes. Two software *i.e.,* CNCI (version 2) and CPC (version 0.9-r2) (http://cpc.cbi.pku.edu.cn/), with default parameters, were used to assess the protein-coding potential of novel transcripts. The intersection of both non protein-coding potential results was chosen as long non-coding RNAs.

Transcripts abundances were quantified by software StringTie is a reference-based approach. For each transcription region, an FPKM (fragment per kilobase of transcript per million mapped reads) value was calculated to quantify its expression abundance and variations. The coding and non-coding RNA transcripts that were differently expressed were investigated, accordingly. DESeq2 was then used to compare the differential expression between two groups (or by edgeR between two samples). The genes/transcripts with the parameter of false discovery rate (FDR) below 0.05 and absolute fold change $\geq 2$ were considered as differentially expressed genes/transcripts.

## Real-time qRT-PCR validation

For expressions quantification, total RNA from spinal cord samples was extracted by using TRIzol reagent (Pufei biological, USA). Extracted RNA was converted into cDNA using a reverse transcription kit (Vazyme, Nanjing, China) as per the instructions of the manufacturer. Internal control for the relative expressions of the gene was set as

**Table 1  qRT-PCR primer sequences.**

|  | Forward primer | Reverse primer |
|---|---|---|
| BDNF | TGATGCTCAGCAGTCAA | CACTCGCTAATACTGTCAC |
| TNF-α | CGGAAAGCATGATCCGAGAT | AGACAGAAGAGCGTGGTGGC |
| IL-1β | TTCAAATCTCACAGCAGCAT | CACGGGCAAGACATAGGTAG |
| IL-6 | AGCCACTGCCTTCCCTAC | TTGCCATTGCACAACTCTT |
| GAPDH | CCTCCTGCACCACCAACTGCTT | GAGGGGCCATCCACAGTCTTCT |

glyceraldehyde-3-phosphate dehydrogenase gene (GAPDH). Relative levels of gene expression were calculated using the 2-$\Delta\Delta Ct$ method ($n = 3$). The primers used are given in Table 1.

## Constructing the CeRNA network

For evaluation DELs-DEMs interactions the starBase database (version 2.0; https://starbase.sysu.edu.cn/starbase2/) was used (*Li et al., 2014*), integrating with miRNA-mRNA interactions to create networking of DEL-DEM-DEG ceRNA employing Cytoscape software (version 3.6.1; http://www.cytoscape.org). Similarly, the DECs and DEMs interactions were coupled and then combined with the miRNA-mRNA interactions using Cytoscape software for creating the network of DEC-DEM-DEG ceRNA.

## Functional enrichment and annotation

Gene ontology (GO) analysis is a popular bioinformatics tool for annotating and analyzing transcriptome data. KEGG is a database that helps researchers better understand the biological system's advanced functions and applications. The data acquired from GO and KEGG pathway database analysis for DEGs were processed by using the Database for Annotation, Visualization, and Integrated Discovery (DAVID, https://david.ncifcrf.gov/). For the genome study, REACTOME (a pathway database) was employed to illustrate the molecular pathway information.

## Statistical analysis

To conduct statistical analysis, we used the GraphPad Prism software. The means ± standard deviations (mean ± SEM) were used to express all of the data. For RT-PCR, three technical replicates were collected from each independent experiment to account for variability in the sample. The significances among multiple groups were determined by One-way Analysis of Variance (ANOVA) followed by Bonferroni post hoc test. $P$ value < 0.05 was considered statistically significant.

## RESULTS

### The suppression of the post-SCI inflammatory response by physical exercise therapy

Rats were divided into three groups: (i) Sham; (ii) SCI; (iii) SCI plus TMT (TMT). Firstly, we successfully constructed an SCI model (Fig. S1). In previous studies, SCI-induced microglial activation and successive release of inflammatory factors *i.e.,* tumor necrosis

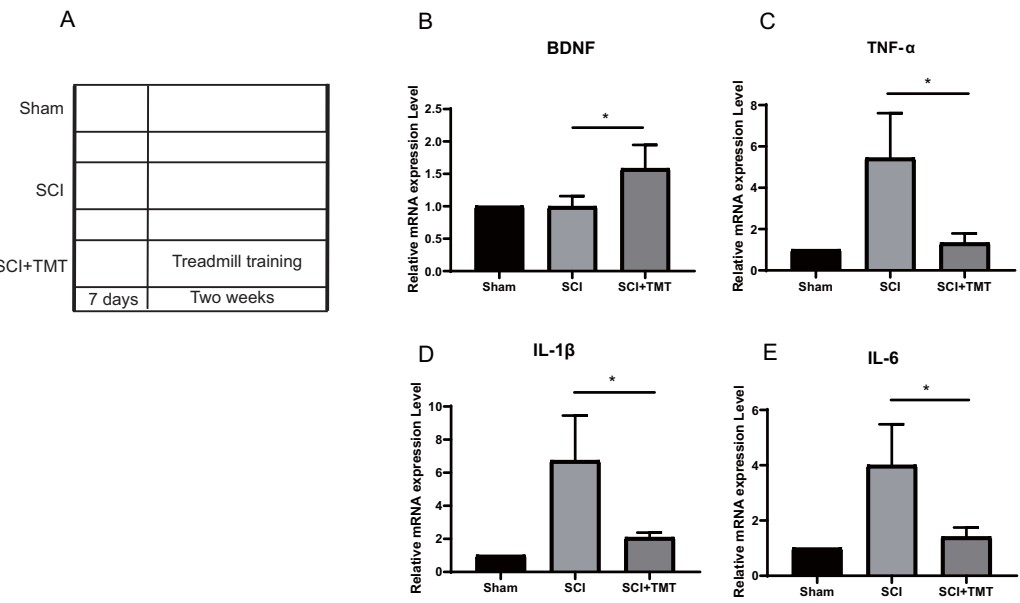

**Figure 1** **The inhibitory effect of exercise therapy on inflammatory response after SCI.** (A) The experimental procedures of the dual rat spinal lesion model and the time course in all rats are shown. TT, treadmill training, treadmill training were performed on rat receiving SCI after 7 days. (B–E) RT-PCR analyses ($n = 3$) of BDNF ($P = 0.0476$), TNF-$\alpha$ ($P = 0.0181$), IL-1$\beta$ ($P = 0.0263$), IL-6 ($P = 0.0265$) in the tissue of lesion region were performed two weeks after treadmill training. Data are expressed as mean $\pm$ SEM. An asterisk (*) indicates $P < 0.05$ *versus* the SCI treatment groups by one-way ANOVA.

factor (TNF), interferon (INF), and interleukin (IL) have been shown to cause direct death of neurons however induction in vascular endothelial cells for the expression of a range of chemotaxis and cell adhesion molecules (*Li et al., 2020a*; *Li et al., 2020b*; *Li et al., 2020c*; *Orr & Gensel, 2018*). In our study, within three weeks of post-SCI, the levels of mRNA expressions of IL-6, TNF-α, and IL-1β were increased. Treadmill training was performed after one week of SCI to assess the anti-inflammation potential of physical exercise therapy. Two weeks later, the levels of mRNA of the three pro-inflammatory cytokines at the site of the lesion were examined (Fig. 1A). All three cytokines were markedly suppressed after treatment with physical activity (Figs. 1B–1E).

## Differential expression analysis

As per the pre-set threshold (FDR value <0.05 and |FC| ≥ 2), we have identified a whole of 76 differential lncRNAs between SCI as well as in control samples, comprising 41 upregulated and 35 downregulated lncRNAs. Whereas 33 differential miRNAs were identified, containing 21 up-regulated and 12 downregulated miRNAs. In the account of mRNAs, 19 differential mRNAs were generated, comprising 11 upregulated and eight downregulated genes. All differential LncRNAs, miRNAs, and mRNAs are presented in Fig. 2 and Tables 2–4.

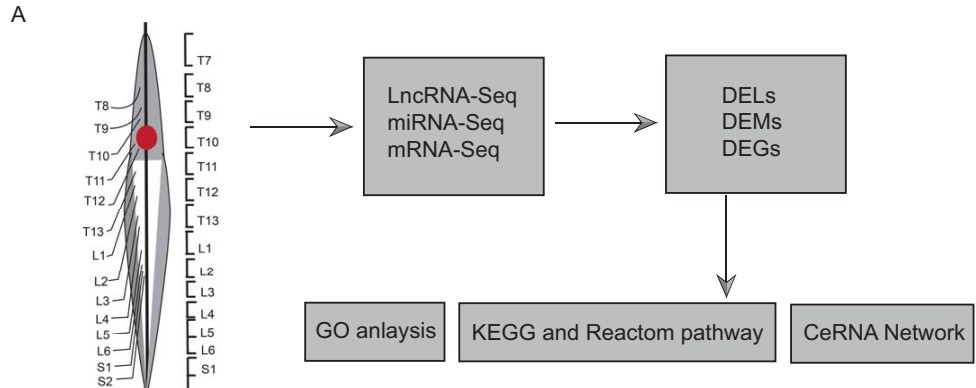

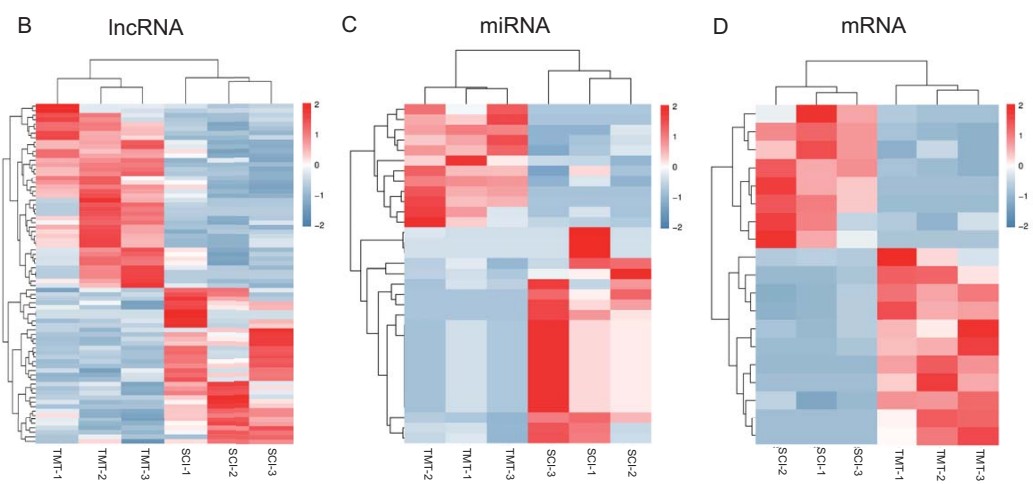

**Figure 2  Hierarchical clustering and heat map analysis of differentially expressed, Differentially expressed genes (DEGs) determined by RNA-seq.** (A) The workflow of RNA-seq. (B) lncRNA, (C) miRNA and (D) mRNA in spinal cord injury. $N = 3$ rats for the SCI groups (SCI) *versus* SCI plus TMT groups (TMT).

## GO ontology

Gene ontology was used to determine which molecular functions and biological processes were overexpressed in DEGs in different analyses employing DAVID 6.8. The top 20 GO terms were shown in Table 4. biological process and 12 cellular components were significantly ($p < 0.05$) identified in "spinal cord injury rats *vs* Control". Most of these processes were related to cellular components, for example, Intracellular organelle, whole membrane, synapse, neuron projection, mitochondrion, macromolecular complex, organelle membrane (Table 5). The DEGs are mainly involved in eight molecular functions, 13 cellular components, 21 biological processes (Figs. 3–4). In the pathophysiology of SCI, these functions are critical for the control of differently expressed mRNAs.

**Table 2  LncRNAs identified from spinal cord injury rats and control.**

| LncRNA | logFC | *P*-value |
|---|---|---|
| MSTRG.33.2 | 10.0812615 | 0.021022906 |
| ENSRNOT00000076320 | 8.79224803 | 0.0107825168758771 |
| MSTRG.11112.11 | 7.29117069932186 | 0.035352209 |
| ENSRNOT00000086366 | 7.129283017 | 0.0114471563265022 |
| ENSRNOT00000083950 | 5.824428435 | 0.0414770168679901 |
| ENSRNOT00000086620 | 3.432959407 | 0.0497646771698065 |
| ENSRNOT00000086821 | 2.669851398 | 0.0186267039307323 |
| ENSRNOT00000086178 | 2.427171255 | 0.00690232969338495 |
| MSTRG.10986.1 | 2.412693559 | 0.000160154329580305 |
| ENSRNOT00000086546 | 1.70160742 | 0.0152804128373893 |
| MSTRG.4137.1 | 1.686633919 | 0.0242778601066535 |
| ENSRNOT00000090814 | 1.647118977 | 0.0488143840391992 |
| MSTRG.7062.1 | 1.51786465 | 0.00793127349116241 |
| MSTRG.3142.3 | 1.300233024 | 0.01383210401143 |
| MSTRG.1588.1 | 1.134194142 | 0.000179570539463743 |
| MSTRG.8085.2 | 1.132755209 | 0.000168225875882159 |
| MSTRG.19821.1 | 1.113138975 | 0.0164053955435965 |
| MSTRG.13512.1 | 1.087462841 | 0.0353603256492267 |
| MSTRG.8083.1 | 1.014592902 | 0.000319423897631703 |
| MSTRG.2838.2 | 0.898338025 | 0.0052426927295388 |
| ENSRNOT00000080918 | 0.87774425 | 0.021320614155435 |
| MSTRG.1747.1 | 0.852631856 | 0.0177697087990876 |
| MSTRG.17839.2 | 0.842643672 | 0.0213202516484291 |
| MSTRG.14369.2 | 0.812629864 | 0.0326086535510531 |
| MSTRG.12598.1 | 0.809435232 | 0.0415891682203027 |
| ENSRNOT00000076871 | 0.786495575 | 0.0326391921614027 |
| MSTRG.19048.1 | 0.739999654 | 0.00828216258847988 |
| ENSRNOT00000081763 | 0.692490965 | 0.03818921639222 |
| MSTRG.1451.1 | 0.682359264 | 0.0394318326751441 |
| MSTRG.14094.1 | 0.648657176 | 0.0255286110499475 |
| MSTRG.18736.1 | 0.647118977 | 0.017146627923167 |
| MSTRG.17413.3 | 0.644264249 | 0.00104354162803263 |
| MSTRG.4152.1 | 0.619535649 | 0.00530706474822808 |
| MSTRG.9250.1 | 0.616196086 | 0.0229040704386873 |
| MSTRG.1989.1 | 0.612171348 | 0.000356634083855439 |
| MSTRG.11374.2 | 0.607763881 | 0.0149094189231467 |
| ENSRNOT00000075902 | 0.604684642 | 0.0166475071815435 |
| ENSRNOT00000076000 | 0.604684642 | 0.0166475071815435 |
| MSTRG.1989.3 | 0.590292661 | 0.00215076264719526 |
| MSTRG.16014.1 | 0.58969731 | 0.0228027576389219 |
| ENSRNOT00000089399 | 0.586862034 | 0.0370607586062419 |
| MSTRG.3891.3 | −0.599156382 | 0.0328769617637782 |

**Table 2** (*continued*)

| LncRNA | logFC | *P*-value |
|---|---|---|
| MSTRG.9743.3 | −0.619030082 | 0.016678145936339 |
| MSTRG.15844.1 | −0.666289706 | 0.00202253984061765 |
| ENSRNOT00000015084 | −0.67215706 | 0.0187497474410401 |
| MSTRG.4710.1 | −0.674713027 | 0.00648103109525139 |
| MSTRG.19470.3 | −0.841476083 | 0.00266923138292542 |
| MSTRG.1989.7 | −0.853536791 | 0.0213306203276205 |
| ENSRNOT00000075969 | −0.929873914 | 0.0347218975818039 |
| ENSRNOT00000076019 | −0.959274062 | 0.00432485597221968 |
| MSTRG.13292.3 | −0.979268558 | 0.000115364004801675 |
| ENSRNOT00000092526 | −1.064130337 | 0.0305839913259358 |
| MSTRG.4415.1 | −1.095157233 | 0.027189419965886 |
| MSTRG.4415.2 | −1.175367087 | 0.0452097190879338 |
| ENSRNOT00000078019 | −1.181169759 | 0.0410096018905549 |
| MSTRG.13261.2 | −1.249290905 | 0.0375332186447193 |
| ENSRNOT00000076087 | −1.367855016 | 0.00169667936735914 |
| MSTRG.11223.1 | −1.606495662 | 0.008217855515345 |
| MSTRG.313.1 | −1.635484276 | 0.0235444573563792 |
| MSTRG.14979.1 | −1.705067671 | 0.0156468198812112 |
| MSTRG.9237.1 | −1.719612027 | 0.0205282663301104 |
| MSTRG.1882.1 | −1.883541023 | 0.00822547540240806 |
| ENSRNOT00000076867 | −2.008112646 | 0.0291130486628121 |
| MSTRG.9743.2 | −2.137503524 | 0.011571659606815 |
| ENSRNOT00000082061 | −2.525461489 | 0.0149733521486662 |
| MSTRG.17976.1 | −2.857165222 | 0.0000002693454398845 |
| MSTRG.12996.1 | −2.926386771 | 0.0205817708129409 |
| MSTRG.19204.13 | −3.688158775 | 0.045320637572825 |
| ENSRNOT00000076212 | −5.400879436 | 0.0426997324624349 |
| MSTRG.9250.2 | −7.351675438 | 0.0384037713653725 |
| ENSRNOT00000076843 | −7.437405312 | 0.0466458729665146 |
| MSTRG.2483.2 | −8.196397213 | $3.60412557930323E-07$ |
| MSTRG.18910.6 | −8.276124405 | 0.0277830513033169 |
| ENSRNOT00000085934 | −8.667702932 | 0.0242142523101195 |
| MSTRG.17772.3 | −9.299208018 | 0.0000000155113930641 |
| ENSRNOT00000086103 | −9.731319031 | 0.0387368354233703 |

## KEGG pathway enrichment analysis of the DEGs

The pathways related to DEGs in SCI rats were mostly enriched, according to KEGG analysis. The differentially identified pathways include metabolic pathways, proteoglycans in cancer, phagosomes, cell adhesion molecules, fluid shear stress, and atherosclerosis, ubiquitin mediated proteolysis, cardiac muscle contraction, ECM-receptor interaction, p53 signaling pathway, lysine degradation, SNARE interactions in vesicular transport, circadian rhythm, and ribosome. The fold enrichment of DEGs involved in each pathway was shown by a bar chart(Fig. 5 and Tables 6–7). According to reactome pathway enrichment analyses,

**Table 3  miRNAs identified from spinal cord injury rats and control.**

| miRNA | LogFC | *P*-value |
|---|---|---|
| miR-6783-x | 6.635125566 | 0.007439362 |
| miR-11987-x | 6.002852512 | 0.028075971 |
| miR-4695-y | 5.173393982 | 0.00894195 |
| miR-206-y | 4.919975631 | 0.025565172 |
| novel-m0155-5p | 4.91041288 | 0.019623 |
| miR-8117-y | 4.170508397 | 0.031183564 |
| miR-4443-x | 3.003103193 | 0.000152928 |
| miR-188-x | 2.026938811 | 0.024781164 |
| miR-3969-x | 1.977625438 | 0.041063373 |
| miR-4510-x | 1.44232092 | 0.013940822 |
| novel-m0090-5p | 1.250712276 | 0.049987453 |
| novel-m0094-5p | 1.250712276 | 0.049795445 |
| novel-m0095-5p | 1.250712276 | 0.049794643 |
| novel-m0097-5p | 1.250712276 | 0.04947851 |
| novel-m0099-5p | 1.250712276 | 0.049489535 |
| novel-m0102-5p | 1.250712276 | 0.049638802 |
| novel-m0106-5p | 1.250712276 | 0.049396397 |
| novel-m0109-5p | 1.250712276 | 0.049109307 |
| novel-m0172-3p | 1.250712276 | 0.048731876 |
| rno-miR-1-3p | 1.240137998 | 5.09E−05 |
| rno-miR-206-3p | 1.130015606 | 0.000595429 |
| miR-6325-y | −1.00206838 | 0.033431201 |
| miR-346-x | −1.540841632 | 0.032325722 |
| miR-12135-y | −1.711770186 | 0.00398775 |
| miR-410-5p | −1.975888032 | 0.010346566 |
| miR-329-x | −2.147890578 | 0.009034055 |
| miR-1224-y | −2.816357195 | 0.046707435 |
| miR-487b-5p | −2.827987993 | 0.030780316 |
| miR-325-x | −3.209453366 | 0.011362352 |
| miR-421-x | −5.014504078 | 0.027900932 |
| miR-193b-5p | −5.205939832 | 0.01268394 |
| miR-381-5p | −5.326009951 | 0.009075182 |
| novel-m0072-3p | −5.478108949 | 0.003935518 |

PSGs bind proteoglycans, and TGF-beta1 was the most significantly affected phase in SCI. This was supported by the findings of the GO enrichment analysis.

## ceRNA network

Nine DEMs were projected to control 45 DELs using the starBase database, and this information was applied for the development of the lncRNA-miRNA-mRNA ceRNA network, which was then interconnected to the miRNA-mRNA network. In this network, there were 73 nodes (20 DEMs; 45 DELs; 8 DEGs), and 246 interactions (21 DEL-DEM and 225 DEM-DEG interactions) (Fig. 6). Particularly, endosome-related signaling

**Table 4  mRNAs identified from spinal cord injury rats and control.**

| mRNA | Gene | LogFC | P-value | FDR |
|---|---|---|---|---|
| ENSRNOG00000000879 | Slc9a6 | 11.67065625 | 8.88E−27 | 5.82E−16 |
| ENSRNOG00000050864 | LOC100910990 | 9.366322214 | 7.13E−20 | 1.61E−06 |
| ENSRNOG00000010214 | Scrn2 | 8.582455645 | 1.30E−10 | 0.044053341 |
| ENSRNOG00000015576 | Hsdl1 | 8.321928095 | 4.75E−05 | 0.043050844 |
| ENSRNOG00000009253 | Igsf9b | 5.257387843 | 5.45E−05 | 0.044410478 |
| ENSRNOG00000010018 | Clec4a3 | 4.486012218 | 4.02E−06 | 0.004862266 |
| ENSRNOG00000014027 | RGD1304728 | 2.496425826 | 2.67E−07 | 0.000543892 |
| ENSRNOG00000014297 | Sdc4 | 2.130703692 | 6.82E−08 | 0.000185403 |
| ENSRNOG00000042492 | Cop1 | 1.175309633 | 1.30E−10 | 7.08E−07 |
| ENSRNOG00000009329 | Nr1d1 | 1.042908591 | 4.25E−07 | 0.00068856 |
| ENSRNOG00000015670 | Stx7 | 0.921466597 | 4.17E−06 | 0.004862266 |
| ENSRNOG00000003392 | Grsf1 | −1.610993791 | 3.49E−05 | 0.035612944 |
| ENSRNOG00000019099 | AABR07054578.1 | −1.800468536 | 2.29E−07 | 0.000533957 |
| ENSRNOG00000008249 | Brms1l | −1.941017121 | 4.64E−07 | 0.00068856 |
| ENSRNOG00000009411 | Chn2 | −3.732716121 | 3.77E−07 | 0.000682447 |
| ENSRNOG00000039494 | Aass | −5.240314329 | 8.73E−06 | 0.009492712 |
| ENSRNOG00000005975 | LOC100362027 | −7.93092503 | 8.88E−27 | 1.45E−22 |
| ENSRNOG00000006357 | Sgip1 | −8.965784285 | 5.03E−08 | 0.000163971 |
| ENSRNOG00000002070 | Mrpl1 | −10.14635653 | 1.06E−06 | 0.001442195 |

pathways were considerably enriched in the genes functional analysis in the network of lncRNA-associated ceRNA (Table 8). The detail of ceRNA network show in Table S4.

## DISCUSSION

In current findings, we investigated that the anti-inflammatory effects of physical exercise therapy on the SCI rats, indicating that physical exercise may play the endogenous protection to spinal cord injury in the rats. In addition, the induction of neurotrophins with exercise, especially BDNF, was also confirmed by our data. Physical exercise is an active area of research that is primarily focused on promoting regeneration after disease or injury. Therefore, our research may shed light on the potential of exercise as a therapeutic intervention for SCI. Next, we demonstrated that the mechanism of physical exercise-induced endogenous protection involves anti-inflammatory and regeneration activities.

Recently, genome-wide association studies provide a new perspective on the pathophysiology of diseases at the molecular level (Ma et al., 2020). Noncoding RNAs (ncRNAs) are becoming more widely recognized as gene regulators (Kopp & Mendell, 2018). According to Zhu et al. (2019) these newly discovered ceRNA interaction axes could

**Table 5  Gene ontology analysis of DEGs in spinal cord injury rats.**

| ID | Descrption | *P*-value | Genes |
|---|---|---|---|
| GO:0007267 | Cell–cell signaling | 0.008695682 | Igsf9b, Nr1d1, Sdc4, Stx7 |
| GO:0043229 | Intracellular organelle | 0.002179334 | Slc9a6, Mrpl1, Grsf1, LOC100362027, LOC100910990 |
| GO:0044424 | Intracellular part | 0.0153761268449414 | Slc9a6, Mrpl1, Grsf1, LOC100362027, Sgip1, Brms1l, Igsf9b, Nr1d1, RGD1304728, Sdc4, Hsdl1, Stx7, Aass, Cop1, LOC100910990 |
| GO:0005737 | Cytoplasm | 0.012045213 | Slc9a6, Mrpl1, Grsf1, LOC100362027, Sgip1, Nr1d1, RGD1304728, Sdc4, Hsdl1, Stx7, Aass, Cop1, LOC100910990 |
| GO:0044422 | Organelle part | 0.015408745 | Slc9a6, Mrpl1, Grsf1, LOC100362027, Sgip1, Brms1l, Nr1d1, RGD1304728, Sdc4, Stx7, Cop1 |
| GO:0043228 | Non-membrane-bounded organelle | 0.012180255 | Mrpl1, Grsf1, LOC100362027, Brms1l, Igsf9b, Nr1d1, RGD1304728, Sdc4 |
| GO:0043232 | Intracellular non-membrane-bounded organelle | 0.012180255 | Mrpl1, Grsf1, LOC100362027, Brms1l, Igsf9b, Nr1d1, RGD1304728, Sdc4 |
| GO:0098805 | Whole membrane | 0.009392774 | Slc9a6, Sgip1, Igsf9b, Sdc4, Stx7 |
| GO:0045202 | Synapse | 0.007481628 | Slc9a6, Igsf9b, Nr1d1, Stx7 |
| GO:0043005 | Neuron projection | 0.010686377 | Slc9a6, Igsf9b, Nr1d1, Stx7 |
| GO:0005622 | Intracellular | 0.01859359 | Slc9a6, Mrpl1, Grsf1, LOC1003620 |
|  |  |  | 27, Sgip1, Brms1l, Igsf9b, Nr1d1, RGD1304728, Sdc4, Hsdl1, Stx7, Aass, Cop1, LOC100910990 |
| GO:0044444 | Cytoplasmic part | 0.020377852 | Slc9a6, Mrpl1, Grsf1, LOC100362027, Sgip1, RGD1304728, Sdc4, Hsdl1, Stx7, Aass, Cop1 |
| GO:0005739 | Mitochondrion | 0.025294767 | Mrpl1, Grsf1, Hsdl1, Aass |
| GO:0097458 | Neuron part | 0.029462285 | Slc9a6, Igsf9b, Nr1d1, Stx7 |
| GO:0044446 | Intracellular organelle part | 0.039921176 | Slc9a6, Mrpl1, Grsf1, LOC100362027, Sgip1, Brms1l, Nr1d1, RGD1304728, Stx7, Cop1 |
| GO:0098588 | Bounding membrane of organelle | 0.040810661 | Slc9a6, Sgip1, Stx7, Cop1 |
| GO:0032991 | Macromolecular complex | 0.064570235 | Mrpl1, Grsf1, LOC10036202, Sgip1, Brms1l, Nr1d1, Stx7, Cop1 |
| GO:0042995 | Cell projection | 0.064149656 | Slc9a6, Igsf9b, Nr1d1, Stx7 |
| GO:0031090 | Organelle membrane | 0.103768137 | Slc9a6, Sgip1, Stx7, Cop1 |

be an important targets for treating intervertebral disc degeneration, including Metastasis-associated lung adenocarcinoma transcript 1 (MALAT1)/hsa_circRNA_102348-hsa-miR-185-5p-TGFB1/FOS,MALAT1-hsa-miR-155-5p-HIF1A, hsa_circRNA_102399-hsa-miR-302a-3p-HIF1A, MALAT1-hsa-miR-519d-3p-MAPK1, and hsa_circRNA_100086-hsa-miR-509-3p-MAPK1 ceRNA axes. After rigorous selection, *Ma et al. (2020)* found that the lncRNA-associated networks of ceRNA in the AD mouse model were revealed to be primarily engaged in memory (Akap5), synaptic plasticity, and regulation of amyloid-β (Aβ)-induced neuro-inflammation (Klf4). The ncRNAs have a critical role in the physiology and pathophysiology of SCI. They are also well-known as a promising candidate for disease biomarkers and therapies (*Gong, Liu & Shen, 2021*). In this study, RNA-seq was employed for systematically analyzing the profiles of mRNA, lncRNA, and miRNA in SCI rats after physical exercise therapy at two weeks post-SCI. In the current study, the SCI rat model

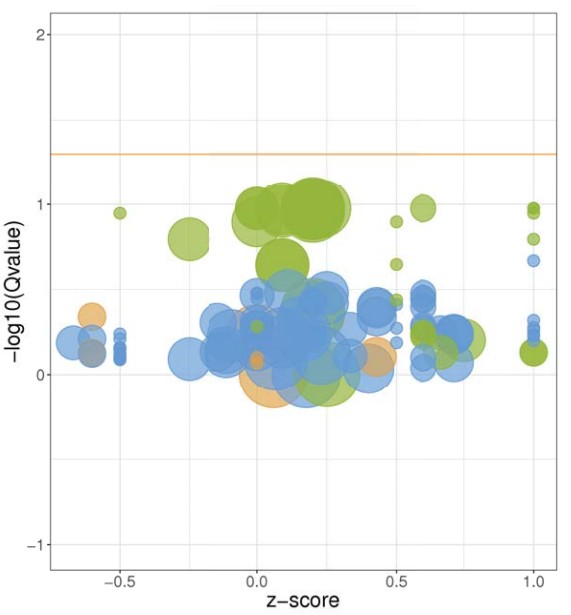

Figure 3 **Enriched GO functional categories.** GO terms are visualized in semantic similarity-based scatterplots. Bubble color indicates the different categories. Bubble size shows how much the GO term is represented in the GO database (right), Selected top 20 categories are shown on the left.

was used for T10 lateral hemisection and exercise treatment was employed for 14 days after SCI operation. Finally, the samples of T10 spinal cord segments were collected for RNA-seq.

The genes in the DEGs were analyzed using GO enrichment and KEGG analysis and we recognized not only several enriched terms involved in cell part including intracellular part, organelle part, cytoplasm, whole membrane, the bounding membrane of organelle, but also various pathways relevant to neuronal rewiring including synapse, neuron projection, neuron part. According to reports, exercise promotes neuronal plasticity and stimulates neurogenesis in the adult CNS (*Wu et al., 2016*; *Horowitz et al., 2020*). KEGG analysis illuminates the key pathways including metabolic pathways and the p53 signaling pathway, accordingly. Exercise regulates the expression of a multitude of genes in the CNS, including neutrotrophins and genes involved in neuronal plasticity, the immune response, and cell death (*Jang et al., 2019*; *Dupont-Versteegden et al., 2004*). As a result, some modulatory molecules acquired from the spinal cord may be responsible for the anti-inflammatory effects of exercise therapy and increased BDNF expression shown in our study. After a traumatic injury, we detected significant changes in the expression of associated ncRNAs and mRNAs in spinal cord tissue, and we projected the structure and possible function of the differentially expressed ncRNAs and mRNAs regulation network. In general, lncRNA and miRNA molecules can have a role in SCI regulation. The LncRNAs and protein-coding mRNAs both act as ceRNAs and super-sponges controlling the expression of miRNA. As a

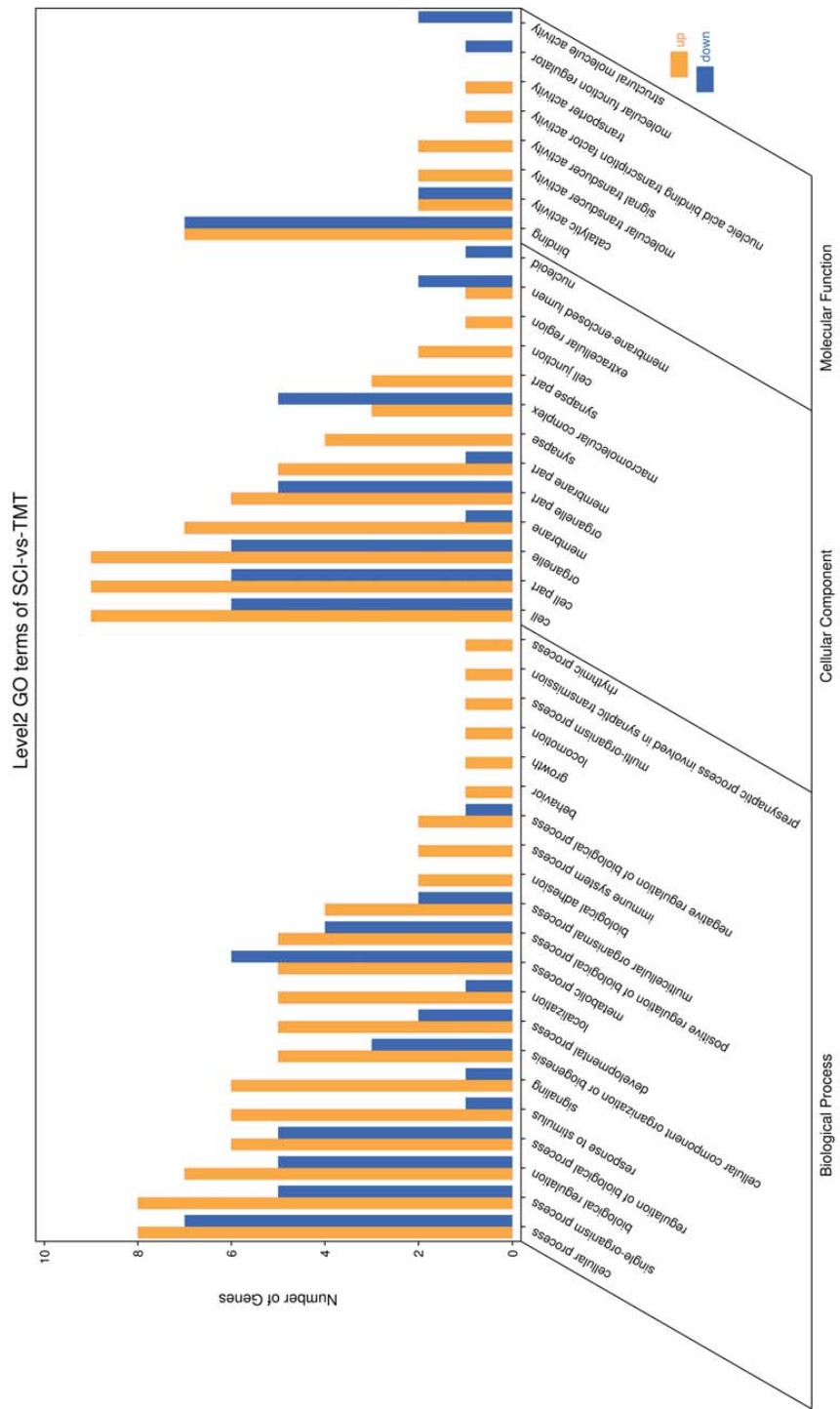

**Figure 4  Enriched GO functional categories.** GO enrichment analysis shows the numbers of genes in different GO term.

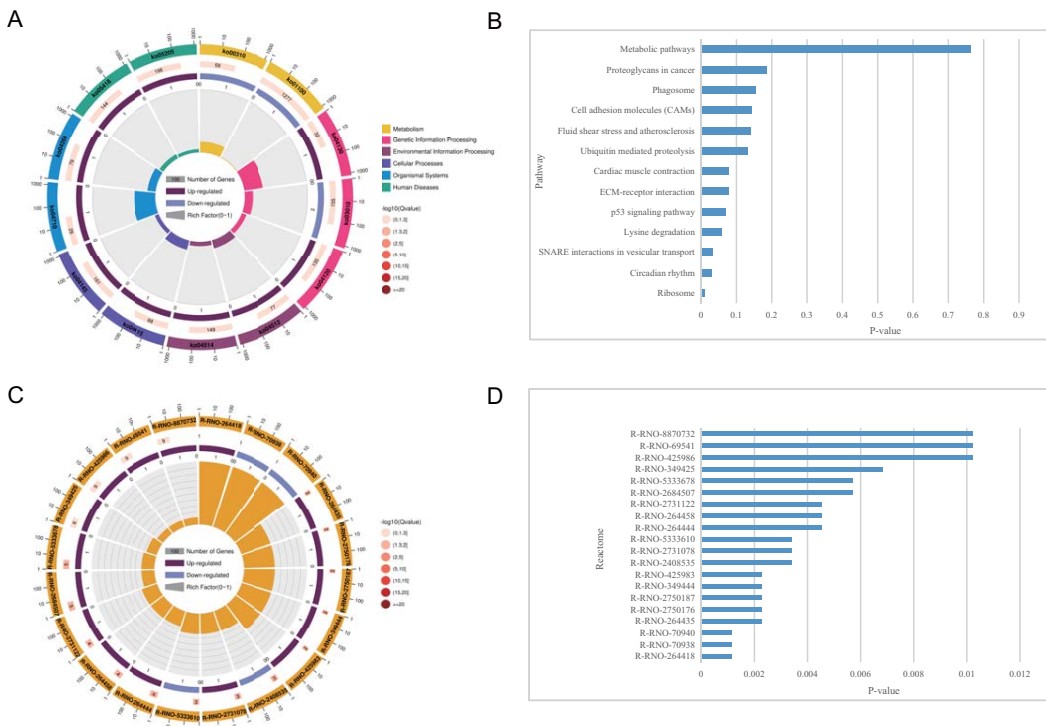

**Figure 5** **KEGG pathway enrichment and reactome pathway enrichment analysis of DEGs.** (A) Circle plot demonstrating the KEGG pathway enriched by the DEGs. Inner plot color corresponds to the rich factor. The second ring displays up and down genes. The third ring displays Q-values. The outer ring displays GO terms. (B) Statistically significant pathways listed and their colors are shown by *P*-value. (C) Circle plot demonstrating the reactome pathway enriched by the DEGs. (D) Top 20 statistically significant pathways listed and their colors are shown by *P*-value.

result, we applied miRanda to predict interactions between miRNA–mRNA and miRNA–lncRNA and established triple networks among DElncRNA–DEmiRNA–DEmRNA for SCI in rats receiving physical exercise therapy. The identified networks of lncRNA-associated ceRNA may help to understand further the effects of exercise on SCI and develop new treatments for the disease.

The data analysis showed different lncRNA-associated ceRNA networks taking part in molecular pathways that may improve recovery in the SCI population through exercise. Syntaxin-7(Stx7) is a syntaxin family membrane receptor involved in vesicle transport (*Wang, Frelin & Pevsner, 1997*). Syntaxin is involved in vesicle docking and fusion, both of which are required for neurotransmitter release (*Hammarlund et al., 2007*; *Li et al., 2020a*; *Li et al., 2020b*; *Li et al., 2020c*). *Ying et al. (2012)* reported that spinal cord can decrease the levels of Syntaxin-3 (Stx3) after traumatic brain injury, which affects membrane homeostasis. The majority of spinal cord vulnerability investigation has concentrated on the effects of syntaxins such as Stx3 and Stx1 (*Matsuzawa et al., 2014*; *Holly et al., 2012*). However, there has been little research into the effects of Stx7 on the spinal cord. The ceRNAs and the gene Stx7 are part of one of these networks,

**Table 6   KEGG pathway of DEGs.**

| Pathway | P-value | Genes |
|---|---|---|
| Ribosome | 0.01025381 | Mrpl1, LOC100362027 |
| Circadian rhythm | 0.02951816 | Nr1d1 |
| SNARE interactions in vesicular transport | 0.03252779 | Stx7 |
| Lysine degradation | 0.05924911 | Aass |
| p53 signaling pathway | 0.06801154 | Cop1 |
| ECM-receptor interaction | 0.07670248 | Sdc4 |
| Cardiac muscle contraction | 0.07862414 | Slc9a6 |
| Ubiquitin mediated proteolysis | 0.1310292 | Cop1 |
| Fluid shear stress and atherosclerosis | 0.1392034 | Sdc4 |
| Cell adhesion molecules (CAMs) | 0.1437155 | Sdc4 |
| Phagosome | 0.1544602 | Stx7 |
| Proteoglycans in cancer | 0.1868507 | Sdc4 |
| Metabolic pathways | 0.7628114 | Aass |

**Table 7   Reactome pathway of DEGs.**

| Reactome | Reactome_Name | P-value | Genes |
|---|---|---|---|
| R-RNO-264418 | Translocation of COP1 from the nucleus to the cytoplasm | 0.001140487 | Cop1 |
| R-RNO-70938 | lysine + alpha-ketoglutarate + NADPH+ H+ => saccharopine + NADP+ + H2O | 0.001140487 | Aass |
| R-RNO-70940 | saccharopine + NAD+ + H2O => alpha-aminoadipic semialdehyde + glutamate + NADH+ H+ | 0.037331594 | Aass |
| R-RNO-264435 | Dissociation of the COP1-p53 complex | 0.002279792 | Cop1 |
| R-RNO-2750176 | Syndecan-4 binds Actn1 | 0.002279792 | Sdc4 |
| R-RNO-2750187 | Syndecan-4:PI(4,5)P2 binds PKC alpha:DAG | 0.002279792 | Sdc4 |
| R-RNO-349444 | Phosphorylation of COP1 at Ser-387 by ATM | 0.037331594 | Cop1 |
| R-RNO-425983 | SLC9A6,7 exchange Na+ for H+ across the early endosome membrane | 0.037331594 | Slc9a6 |
| R-RNO-2408535 | Sec-tRNA(Sec):Eefsec:GTP binds to Rpl30 | 0.040704263 | LOC100362027 |
| R-RNO-2731078 | Syndecans 2, (4) bind TGFB1 | 0.040704263 | Sdc4 |
| R-RNO-5333610 | Rpl30:Met-tRNAi:mRNA:Secisbp2:Sec-tRNA(Sec):Eefsec:GTP is hydrolysed to Rpl30:Met | 0.040704263 | LOC100362027 |
| R-RNO-264444 | Autoubiquitination of phospho-COP1(Ser-387 ) | 0.042620458 | Cop1 |
| R-RNO-264458 | Proteasome mediated degradation of COP1 | 0.042620458 | Cop1 |
| R-RNO-2731122 | Syndecans 1, 2 & 4 bind VTN | 0.042620458 | Sdc4 |
| R-RNO-2684507 | Syndecans 1, 2, 4, (3) bind FGF2 | 0.046591965 | Sdc4 |
| R-RNO-5333678 | CPNEs bind PL | 0.046591965 | LOC100910990 |
| R-RNO-349425 | Autodegradation of the E3 ubiquitin ligase COP1 | 0.006825207 | Cop1 |
| R-RNO-425986 | Sodium/Proton exchangers | 0.010221905 | Slc9a6 |
| R-RNO-69541 | Stabilization of p53 | 0.010221905 | Cop1 |
| R-RNO-8870732 | PSGs bind proteoglycans and TGF-beta1 | 0.010221905 | Sdc4 |

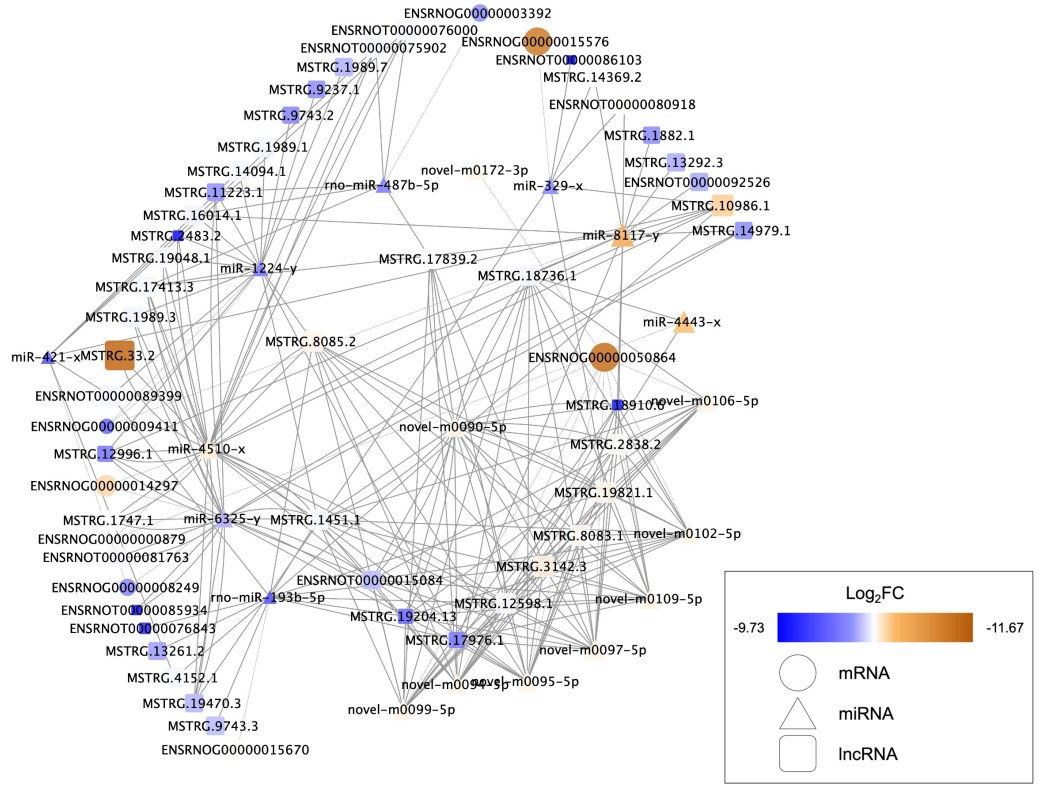

**Figure 6** **Competing endogenous RNA interaction network of lncRNA-miRNA-mRNA.** Blue represents upregulated expression, whereas brown represents downregulated expression. Square nodes represent lncRNAs, triangular nodes represent miRNAs and oval nodes represent mRNAs. Hub genes are indicated by red boxes. FC, fold change; lncRNA, long noncoding RNA; miRNA/miR, microRNA.

**Table 8** **Gene ontology analysis of ceRNA.**

| Enrichment | Count | *P*-value | Genes |
|---|---|---|---|
| Plasma membrane | 4 | 1.7E−1 | Stx7, SLC9A6, Sdc4, Cop1 |
| Membrane | 5 | 2.0E−1 | Stx7, SLC9A6, Sdc4, Cop1, LOC100362027 |
| Transmembrane helix | 3 | 6.6E−1 | Stx7, SLC9A6, Sdc4 |
| Transmembrane | 3 | 6.7E−1 | Stx7, SLC9A6, Sdc4 |
| Integral component of membrane | 3 | 6.9E−1 | Stx7, SLC9A6, Sdc4 |

including LNC ENSRNOT00000089399, LNC MSTRG.13261.2, LNC MSTRG.1451.1, LNC MSTRG.12598.1, LNC MSTRG.19204.13, LNC ENSRNOT00000076843, LNC MSTRG.17976.1, LNC MSTRG.10986.1, LNC MSTRG.8085.2, LNC MSTRG.4152.1, LNC MSTRG.3142.3, LNC MSTRG.19821.1, LNC MSTRG.9743.3. These ceRNAs show the ability for binding miR-193b-5p, targeting the Stx7. NHE6, an endosomal sodium-hydrogen exchanger known to be important in the regulation of endosomal pH, is encoded by Solute carrier family 9 member A6 (SLC9A6) (*Yamashiro et al., 1984*; *Xinhan et al., 2011*). Defects in the SLC9A6/NHE6 transporter have been linked to changes in

somatosensory functions, according to a recent study (*Sikora et al., 2016*; *Petitjean et al., 2020*). Defects in the spinal cord, the nociceptor level, or supraspinal locations can cause sensory impairments. In the dorsal and ventral horns, as well as along the pericentral canal, NHE6/Y KO animals showed enhanced microglial and astrocytic immunoreactivity, indicating that exercise training may increase the levels of SLC9A6/NHE6 to decrease the sensory impairments after SCI. The current study found that exercise training is effective in alleviating the neuropathic pain caused by partial SCI in rats (*Li et al., 2020a*; *Li et al., 2020b*; *Li et al., 2020c*). In this study, it was discovered that SLC9A6 was elevated. MiR-4443, the same as Stx7, has been demonstrated to influence SLC9A6 expression. Furthermore, MiR-4443 could also be sponged by LNC MSTRG.18736.1 and MSTRG.12598.1. Syndecans-4 (SDC4) is a member of the Syndecans (SDCs) family of transmembrane heparan sulphate proteoglycans located on the cell surface (*Leblanc et al., 2018*). Through their glycosaminoglycan chains, SDCs have been stated for interacting with growth factors and extracellular matrix molecules. The effects of SDC4 are achieved as an independent receptor for the platelet-derived growth factors (PDGFs), fibroblast growth factor receptors (FGFR1–FGFR4), vascular endothelial growth factors (VEGFs), as well as fibroblast growth factors (FGFs) (*Elfenbein & Simons, 2013*). However, at present, no study reports the relationship between the SDC4 and BDNF yet. We suspected that SDC4 plays a significant role in the regulation of BDNF. The results of this investigation revealed that miR-6325 may be involved in the regulation of SDC4 expression, whereas MSTRG.17413.3, MSTRG.1989.3, MSTRG.19470.3, MSTRG.11223.1, MSTRG.14979.1, MSTRG.1747.1, ENSRNOT00000015084, MSTRG.12996.1, ENSRNOT00000085934, ENSRNOT00000081763, MSTRG.1451.1, and MSTRG.19204.13 could interact with miR-6325 as a ceRNA.

## CONCLUSIONS

In conclusion, the profiles of lncRNA-associated ceRNA of SCI and physical exercise therapy in rats were explained. The regulatory roles of exercise-induced alterations in gene expression and the present understanding of ceRNA biology have been improved by our findings. We identified Slc9a6, Sdc4, Stx7 as crucial genes in SCI rats with physical exercise therapy, which leads to the hypothesis that exercise promotes efficient signaling and neuronal plasticity in the spinal cord.

## ACKNOWLEDGEMENTS

The authors wish to acknowledge Dr Chunlei Shan, Professor of School of Rehabilitation Science, Shanghai University of Traditional Chinese Medicine, for his help in interpreting the significance of the results of this study.

### Funding

This study was supported with grants from the National Natural Science Foundation of China (82072536), the Natural Science Foundation of Jiangsu Province (BK20191182), the Science and Technology Plan Project of Suzhou (SYS2019016), and the Medical and Health Science Project of Suzhou New District (2018Z008). The funders had no role in study design, data collection and analysis, decision to publish, or preparation of the manuscript.

### Grant Disclosures

The following grant information was disclosed by the authors:
National Natural Science Foundation of China: 82072536.
Natural Science Foundation of Jiangsu Province: BK20191182.
Science and Technology Plan Project of Suzhou: SYS2019016.
Medical and Health Science Project of Suzhou New District: 2018Z008.

### Competing Interests

The authors declare there are no competing interests.

### Author Contributions

- Jiahuan Wu conceived and designed the experiments, performed the experiments, analyzed the data, prepared figures and/or tables, and approved the final draft.
- Xiangzhe Li performed the experiments, analyzed the data, prepared figures and/or tables, and approved the final draft.
- Qinghua Wang performed the experiments, authored or reviewed drafts of the article, and approved the final draft.
- Sheng Wang performed the experiments, authored or reviewed drafts of the article, and approved the final draft.
- Wenhua He performed the experiments, authored or reviewed drafts of the article, and approved the final draft.
- Qinfeng Wu conceived and designed the experiments, authored or reviewed drafts of the article, and approved the final draft.
- Chuanming Dong analyzed the data, authored or reviewed drafts of the article, and approved the final draft.

### Animal Ethics

The following information was supplied relating to ethical approvals (*i.e.*, approving body and any reference numbers):

The Animal Care and Use Committee of Nantong University provided full approval for this research (S20200317-024).

### Data Availability

The data is available at NCBI SRA: SRR17599934, SRR17599935, SRR17599936, SRR17599937, SRR17599938, SRR17599939.

## Supplemental Information

Supplemental information for this article can be found online at http://dx.doi.org/10.7717/peerj.13783#supplemental-information.

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
