# Peer review of "LncRNA/miRNA/mRNA ceRNA network analysis in spinal cord injury rat with physical exercise therapy"

_PeerJ, doi:10.7717/peerj.13783_

## Round 0.1 · original submission · Major Revisions

Thank you for submitting the manuscript to PeerJ. It has been reviewed by experts in the field and we request that you make major revisions before it is processed further.

We look forward to hearing from you soon.

Best wishes,

Badicu Georgian, Ph.D

Reviewer 1 ·

Basic reporting

It is a professional English language used throughout.
The background shows context, it is relevant for the research, but not so more informations about the physical exercise in the context.
The structure conforms to Peerj standards.
Figures are well labelled.

Experimental design

It is an original research and a novelty in field of analysis the role of physical exercises.
It is not clear how is the relationship between mRNA and spinal cord injury. Whys this approach?Please explain.
What means rats functional status?Please explain
The research is rigorous and respect high technical & ethical standard
Methods described have sufficient detail but are some question about line 119-120, `` If multiple arguments are vectors a 120 matrix or array of results is returned. values for alpha and power and…..`seems to lack some informations.
It is not clear the rats groups, it is not a complet description, seems it is a group which has spine section and a group which participate to the training? Or both participate?
Line 130- you speak about 2 groups…which?
Any other inflammatory markers, used for quantifiy the anti-inflamator effects of physical exercise?

Validity of the findings

Anti-inflamatory effects seems to develop after few month of physical exercises. How did you proceed regarding this?
Conclusions are well stated, linked to original research question

Additional comments

4.General comments
The papers is very interesting and well argumentated but a little bit hard to follow and for this reason I suggest to reorganize the results and present more clear the role of physical exercises.

Reviewer 2 ·

Basic reporting

In this study, the author performed RNA-seq and microRNA-seq to examine the differentially expressed circRNAs, lncRNAs, miRNAs, and genes between SCI and TMT groups. The author also established the ceRNAs network using interactions between miRNAs and mRNAs, as well as between miRNAs and lncRNAs/circRNAs. These findings may support that exercise promotes efficient signaling and neuronal plasticity in the spinal cord. Overall, the study is meaningful. I have some minor comments as follows:
The English writing needs to be improved to ensure clear understanding. In the methods part, the writing of the manuscript is not standardized. I suggest the author improve this part.
1) Line 120: repeated punctuation
2) Line 122: “value of .5” abnormal punctuation
3) Line124: “samples sizes” The first word of a sentence should be capitalized.
4) Line 127: “value though https” Here should be “through”.
5) Line 115: “CV” Uncommon abbreviations should be spelled out the first time.
6) Line 85 and Line 116: Please use one space following periods.
7) Line 112: “being compared. within groups” abnormal punctuation here
8) Line 130 “TMTgroup” no space between two words
9) Line 88 and Line 99: “Bladder care”, “all Rats” The capital letter is not necessary here.
10) Line 218: “mode” Here should be “model”.
11) Line 57 and Line 62: The font of in-text citations here is different from other parts.

Experimental design

1) How does the author establish the ceRNAs network? Although the author used the software or database, the interaction is based on what kind of condition? Expression level or correlation coefficient? LncRNAs, miRNAs, and mRNAs, are they positively correlated or negatively correlated?
2) Line 210: “The means ± standard deviations (mean ± SD) were used to express all of the data.” However, in the Figure 1 legend, “Data are expressed as mean ± SEM.” It is not consistent.
3) Figure 2 (B-D): The size of each group name is too small. It is not clear.
4) Line 34: “circRNAs (DECs), lncRNAs (DELs), miRNAs (DEMs), and genes (DEGs) between SCI and normal controls” Here the “normal controls” means “Sham group” or “TMT group”? But in Figure 1 (B-D), the heatmaps showed the gene patterns of SCI groups and TMT groups. Please explain it clearly.

Validity of the findings

no comment

Additional comments

no comment

Reviewer 3 ·

Basic reporting

The manuscript – LncRNA-miRNA-mRNA ceRNA network analysis in Spinal cord injury rat with Physical exercise therapy investigated the effects of exercise therapy on the long noncoding RNA expression. Overall, the background and methods they used were clear and straightforward. Certain RT-PCR of potential inhibitory effect of exercise therapy on inflammatory response are quite significant. They provide a resource of differntitial expressed circRNA , lncRNA miRNA and differential expressed genes between spinal cord injury and normal rates.

Experimental design

The Experimental design and bioinformatic analysis were well defined. They proposed a question how the effect of exercise therapy influence the expression changes on long noncoding RNA, microRNA and mRNA in rate with spinal cord injure. They established a spinal cord injury model and traced the effects. Further lncRNA-seq miRNA-seq and mRNA-seq were perfomed to detected the molecular changes. GO analysis, KEGG and ceRNA network were further analyzed. Addtionally, they performed the qRT-PCR to validate the suppression of the post-SCI inflammatory response to physical exercise therapy.

Validity of the findings

They used the qRT-PCR validation to validate the inhibitory effect of exercise therapy on inflammatory response after SCI.

Additional comments

Minor comments

Figure 1A the figure seemed incomplete

Figure 2: Please describe in detail how your compared the DEG, DELs and DEMs?

Figure 3B The x axis is not clear

Table 4: I am wondering why you have so few DEGs, and why using the Pvalue in the table not FDR described in your method?

---

## Round 0.2 · Minor Revisions

Thank you for submitting the manuscript to PeerJ. Great improvements were performed in the manuscript. Currently, the article is acceptable with minor revisions.

We look forward to hearing from you soon.

Best wishes,

Badicu Georgian, Ph.D

Reviewer 2 ·

Basic reporting

No comments

Experimental design

No comments

Validity of the findings

No comments

Additional comments

All remaining issues were adequately addressed.
There are two minor points about the format of the manuscript.
1. Figure 6. There were no red and green colors here. The colors that I saw were blue and brown.
2. Abnormal space in Line 382 and 383.

---

## Round 0.3 · accepted · Accept

The manuscript is ready to be accepted.